# Data Completion, Model Correction and Enrichment Based on Sparse Identification and Data Assimilation

Daniele Di Lorenzo [1,2,*], Victor Champaney [2,3,*], Claudia Germoso [4], Elias Cueto [5] and Francisco Chinesta [1,2,3,6,*]

1 ESI Group, 3bis Rue Saarinen, CEDEX, 94528 Rungis, France
2 PIMM, CNRS, ENSAM, HESAM, 151 Boulevard de l'Hôpital, 75013 Paris, France
3 ESI Group Chair CREATE-ID, ENSAM Institute of Technology, 151 Boulevard de l'Hôpital, 75013 Paris, France
4 Instituto Tecnologico de Santo Domingo, Av. Los Proceres, Jardines del Norte, Santo Domingo 10602, Dominican Republic; claudia.germoso@intec.edu.do
5 Aragon Institute of Engineering Research, Universidad de Zaragoza, Maria de Luna s/n, 50018 Zaragoza, Spain; ecueto@unizar.es
6 CNRS@CREATE Ltd., 1 Create Way, #08-01 CREATE Tower, a s/n, Singapore 138602, Singapore
* Correspondence: daniele.dilorenzo@esi-group.com (D.D.L.); victor.champaney@ensam.eu (V.C.); francisco.chinesta@ensam.eu (F.C.)

**Abstract:** Many models assumed to be able to predict the response of structural systems fail to efficiently accomplish that purpose because of two main reasons. First, some structures in operation undergo localized damage that degrades their mechanical performances. To reflect this local loss of performance, the stiffness matrix associated with the structure should be locally corrected. Second, the nominal model is sometimes too coarse grained for reflecting all structural details, and consequently, the predictions are expected to deviate from the measurements. In that case, there is no small region of the model that needs to be repaired, but the entire domain needs to be repaired; therefore, the entire structure-stiffness matrix should be corrected. In the present work, we propose a methodology for locally correcting or globally enriching the models from collected data, which is, upon its turn, completed beyond the sensor's location. The proposed techniques consist in the first case of an L1-minimization procedure that, with the support of data, aims at the same time period to detect the damaged zone in the structure and to predict the correct solution. For the global enrichment, instead, the methodology consists of an L2-minimization procedure with the support of measurements. The results obtained showed, for the local problem, a correction up to 90% with respect to the initially incorrectly predicted displacement of the structure, and for the global one, a correction up to 60% was observed (this results concern the problems considered in the present study, but they depend on different factors, such as the number of data used, the geometry or the intensity of the damage). The benefits and potential of such techniques are illustrated on four different problems, showing the large generality and adaptability of the methodology.

**Keywords:** model correction; model enrichment; data completion; SHM; sparse regularization; L1-norm; L2-norm; LASSO; model order reduction

## 1. Introduction

In science and engineering, a model constitutes a linear or non-linear relation between inputs (the actions) and outputs (the reactions). For instance, when considering a deformable solid, the elastic model relates forces and displacements at each point. When the resulting problem cannot be rigorously solved analytically, it is discretized by approximating the involved fields using a number of terms that are sometimes large but always finite, such as N (each one related to a coefficient that must be calculated). When introducing these approximations into the so-called problem's weak form, it results in an algebraic problem of size N (N equations for calculating the N approximation unknowns). In the transient or

nonlinear cases, many algebraic systems must be generally solved in the former case for computing the time evolution of the searched solution and in the last case for ensuring the convergence of the non-linear solution procedure, which operates by linearization.

All this is currently very well known, and it is performed every day in many routine applications of numerical simulation. However, when fast and accurate solutions becomes compulsory, three main difficulties remain open:

1.  The computational costs associated with the solution of those algebraic problems of size $N$, when $N$ becomes large, e.g., several millions. In that case, the real-time feedback compulsory in most of the very timely digital applications is compromised. The use of larger and more powerful computing platforms is not the most appealing alternative due to the fact that online applications in autonomous systems need to embed these solvers in deployed systems in order to monitor the systems for real-time decision-making, based on efficient diagnosis and prognosis.

2.  The second difficulty concerns the underlying model's quality. In some cases, the existing and experienced models are very accurate; however, in many other cases, as the ones concerning very large and complex systems, involving many coupled physical phenomena, seriously compromised the validity of predictions performed with the established state-of-the-art models.

3.  The third difficulty concerns the efficient treatment of parametric models, for which its solutions will depend on the parameters involved in the problem's formulation. Sometimes these parameters concerns the physical model itself; other times, they affect the geometry of the domain in which the problems are defined; finally, at other times, these parameters are related to the loading (in the most general sense) that the system is experiencing. In all cases, when addressing parametric models, the curse of dimensionality (combinatorial explosion) joins the scene. Calculating the parametric solution of the problem could be an excellent opportunity for real-time engineering: for instance, solution particularization, optimization, inverse analysis, uncertainty propagation, and simulation-based control; all are performed under the stringent real-time constraint. However, the curse of dimensionality makes the calculation of those parametric solutions difficult when the number of parameters that they involve increases.

In this general framework, the first issue can be addressed with Model Order Reduction (MOR) techniques [1–4]. The second one can be addressed within an hybrid engineering paradigm, where collected and assimilated data can be used for enriching state-of-the-art models [5,6]. The third issue can be addressed in the context of advanced regressions that are able to address richness (nonlinearities) in multi-parametric settings (many parameters), keeping the amount of required data reasonable and avoiding overfitting phenomena [7,8].

The present work will focus specifically on the second difficulty to address two types of scenarios:

1.  When a model, called a nominal model, is expected to predict the system's response from the applied loading but fails to predict the measurements carried out due to a localized deficiency in it. This is the situation when a model is accurate enough to describe the subjacent physics almost everywhere, except in a small region where the reality differs from the assumed model. This is the case when, in solid mechanics, some localized damage occurs, which involves a degradation of the real mechanical properties with respect to the ones assumed in the so-called nominal model.

2.  When a model constitutes an approximation of the real system everywhere but is not accurate enough everywhere. This is the case when modeling large structures, where the discretization remains poor for reflecting all structural details. Thus, all these details (stiffeners, holes, etc.) are ignored and the resulting model could constitute an approximation (sometimes a crude approximation) of the real system, which is valuable for certain applications but inefficient in other cases.

In the first case,we can talk of damage detection, and over the last decades, many techniques have been developed [9]. In particular, it is possible to classify the general used methods in frequency and time domains. Frequency domain techniques [10–14] are based on modal analysis, and this can be represent a great advantage when a range of frequencies to be analysed is known a priori. Time domain techniques [15–20] are, instead, based on the study of vibration features and signal processing techniques. Wavelet analysis [21–25] is another used technique that uses a wavelet transformation on the modal shape of vibration. Due to its nature, these techniques are largely used for non-destructive testing. Moreover, during the last years, the interest to refine finite element models has grown, and an extensive review is provided in [26,27]. This paper aims to propose a new methodology able to complete the data acquired from few sensors, as well as to find the model's correction (a sort of model reparation). This correction will be used to identify the damage localization and its severity. For that purpose, sparse identification will be the main protagonist within an hybrid approach combining the nominal model with the requested model enrichment, which is the last learned from the assimilated data. The second scenario is even more challenging due to the fact that the model should be corrected (enriched) everywhere and, in general, only a few data available. In this case, since the enrichment does not exhibit localization, the sparse identification will be replaced by global identifications. The next section describes the main methodologies that will be used in the proposed data completion and model enrichment procedures, particularly sparse regularization and identification. Section 3 will focus in the main paper's proposal, the procedures for completing the data and enriching the nominal models in both cases when the enrichment exhibits, or not, localization. Finally, Section 4 will present and discuss some numerical results. In particular, for the local correction procedure, three different geometries (simple plate in Section 4.1, plate with hole in Section 4.2 and an L-shape structure in Section 4.3) are used to show the generality and applicability of the methodology. On the other hand, we only focus on one problem for global corrections (simple plate in Section 4.4).

## 2. Methods: Sparse Regularization and Identification

When addressing parametric problems, the use of regressions is quite normal [8]. We consider, for the sake of simplicity, a simple model $y(\mu)$, with $y$ being the quantity of interest that depends on parameter $\mu$.

If three data are assumed to be known, $\{(\mu_1, y_1), (\mu_2, y_2)$ and $(\mu_3, y_3)\}$, one is tempted to approximate function $y(\mu)$ by using a polynomial basis according to the following.

$$y(\mu) = a_0 + a_1\mu + a_2\mu^2. \tag{1}$$

If we assume that these data are free of noise, it is possible to calculate the three approximation coefficients, $a_0$, $a_1$ and $a_2$, from relation (1) by solving the following system.

$$\begin{pmatrix} 1 & \mu_1 & \mu_1^2 \\ 1 & \mu_2 & \mu_2^2 \\ 1 & \mu_3 & \mu_3^2 \end{pmatrix} \begin{pmatrix} a_0 \\ a_1 \\ a_2 \end{pmatrix} = \begin{pmatrix} y_1 \\ y_2 \\ y_3 \end{pmatrix}. \tag{2}$$

In the case of having $M$ data, the approximation generalizes to the following:

$$y(\mu) = \sum_{k=0}^{M-1} a_k\mu^k, \tag{3}$$

and the procedure for computing the coefficients $a_k$, $k = 0, \ldots, M - 1$ follows a similar procedure. Thus, the approximation richness follows the data's availability. However, we would like to increase the richness without necessarily increasing the amount of data. Imagine, for a while, that the problem solution reads $y(\mu) = \mu^3$. The question is as follows: Can we determine this solution with only three data? Obviously, by using approximation

basis $\{1, \mu, \mu^2\}$, it is not possible. A possibility consists of taking a richer basis, e.g., $\{1, \mu, \mu^2, \mu^3, \mu^4\}$, that, as it can be noticed, contains the solution; however, if only three data are available, the resulting algebraic system

$$
\begin{pmatrix} 1 & \mu_1 & \mu_1^2 & \mu_1^3 & \mu_1^4 \\ 1 & \mu_2 & \mu_2^2 & \mu_2^3 & \mu_2^4 \\ 1 & \mu_3 & \mu_3^2 & \mu_3^3 & \mu_3^4 \end{pmatrix} \begin{pmatrix} a_0 \\ a_1 \\ a_2 \\ a_3 \\ a_4 \end{pmatrix} = \begin{pmatrix} y_1 \\ y_2 \\ y_3 \end{pmatrix},
\tag{4}
$$

is underdetermined (there are infinite solutions). Thus, depending on the strategy followed for solving it (backslash, pseudo-inverses, optimization formulation with different norm choices, etc.), different solutions will be obtained, all them representing perfectly the three data, but they will differ when considering any other value $\mu \neq \{\mu_1, \mu_2, \mu_3\}$.

If an algorithm can provide the right solution, in our case $a_3 = 1$ and $a_0 = a_1 = a_2 = a_4 = 0$, one can consider an extremely rich approximation basis, enabling the approximation of the searched solution by selecting very few terms in it. The amount of data will scale with the number of terms in the rich basis needed for approximating the searched solution instead of scaling with the approximation's basis size.

Thus, when addressing our problem $y(\mu)$, we consider a very rich approximation basis composed of M linearly independent functions $\mathcal{G}_i(\mu)$, $i = 1, \ldots, $M and D data $y_k$, $k = 1, \ldots, $D, with D $\ll$ M, and the resulting algebraic system reads:

$$
\begin{pmatrix} \mathcal{G}_1(\mu_1) & \mathcal{G}_2(\mu_1) & \cdots & \mathcal{G}_{\text{M}}(\mu_1) \\ \mathcal{G}_1(\mu_2) & \mathcal{G}_2(\mu_2) & \cdots & \mathcal{G}_{\text{M}}(\mu_2) \\ \vdots & \vdots & \ddots & \vdots \\ \mathcal{G}_1(\mu_{\text{D}}) & \mathcal{G}_2(\mu_{\text{D}}) & \cdots & \mathcal{G}_{\text{M}}(\mu_{\text{D}}) \end{pmatrix} \begin{pmatrix} a_0 \\ a_1 \\ \vdots \\ a_{\text{M}-1} \end{pmatrix} = \begin{pmatrix} y_1 \\ y_2 \\ \vdots \\ y_{\text{D}} \end{pmatrix},
\tag{5}
$$

or in a more compact form:

$$
\mathbf{Ga} = \mathbf{Y}.
\tag{6}
$$

If it is expected that only N functions (N $\leq$ D $\ll$ M), D $\sim$ N suffices for approximating, with adequate accuracy, solution $y = y(\mu)$; then, only N non-zero coefficients are expected in vector $\mathbf{a}$. Thus, one would prefer solving Equation (6) under the sparsity constraint. It is well known that standard procedures are unable to produce such sparse solutions, as it is the case when using the usual pseudo-inverses or the most common optimization based on the use of the L2-norm, and the last procedure leads to:

$$
\mathbf{a} = \texttt{argmin}_{\mathbf{a}^*} \|\mathbf{Ga}^* - \mathbf{Y}\|_2^2,
\tag{7}
$$

that results in

$$
(\mathbf{G}^T\mathbf{G})\mathbf{a} = \mathbf{G}^T\mathbf{Y},
\tag{8}
$$

for which its solution becomes almost fully populated, rendering the procedure inefficient to account for sparsity.

In order to enforce sparsity, the use of the L0-norm is the best choice; however, its numerical treatment is extremely difficult. For that reason, intermediate norms are considered. The L1-norm has been widely considered and different procedures are used for enforcing it, as was the case for the one used in [7] for performing sparse identification. The so-called *elastic net* regularization combines the *ridge* regularization [28], which alleviates overfitting, with the *lasso* regularization [29] that enforces sparsity. The elastic net regularization is described as follows:

$$
\mathbf{a} = \texttt{argmin}_{\mathbf{a}^*} \left\{ \|\mathbf{Ga}^* - \mathbf{Y}\|_2 + \lambda \left\{ \alpha \|\mathbf{a}^*\|_2^2 + (1 - \alpha) \|\mathbf{a}^*\|_1 \right\} \right\},
\tag{9}
$$

and it allows recovering the usual L2-norm problem when $\lambda = 0$; for $\lambda \neq 0$, the resulting *elastic net* regularized formulation reduces to *ridge* regularization for $\alpha = 1$ and to the *lasso* regularization for $\alpha = 0$. These regularizations were adapted to variable separations for addressing highly multi-parametric models in [8]. The *lasso* regularization was extensively employed in the context of compressed sensing [30,31].

Smaller norms can be also considered, as for example the p-norm, from which the minimization reads as follows:

$$\begin{cases} \min_{\mathbf{a}} \sum\limits_{i=0}^{M-1} |a_i|^p \\ \text{subjected to} : \mathbf{Ga} = \mathbf{Y} \end{cases}, \tag{10}$$

which can be efficiently solved by using the FOCUSS algorithm [32]. In the next section, we show firstly an optimization procedure that uses the L1-norm together with the data when sparsity constrain has to be satisfied. Secondly, the classic L2-norm minimization procedure is presented when the sparsity constraint is no longer required by the problem.

## 3. Data Completion and Models Correction or Enrichment

### 3.1. Model Local Correction and Data Completion

Efficient diagnoses for structural monitoring are of major relevance in most of engineering applications. Different techniques, based on the analysis of the structure response in the static and dynamic regimes, for identifying the existence of damage, as well as for inferring its location and severity, have been extensively considered in numerous recent works. A short review on these techniques is provided for the sake of completeness in Appendix A. In the present section, we will consider an alternative approach that will identify indirectly the damage from the achieved model local correction. The main advantages of the proposed procedure include the ability to locally repair the model, for instance, in presence of localized damage, and its capacity to complete the collected data. For describing the proposed methodology, we consider a structural model represented by the FEM algebraic system $\mathbf{KU} = \mathbf{F}$, where $\mathbf{K}$ is the stiffness matrix, $\mathbf{U}$ is the displacement and $\mathbf{F}$ is the force applied to the structure. We assume that the model's validity was checked in absence of damage. For that purpose, data are collected in a set of points, the sensor's location, corresponding for the sake of simplicity with some nodal positions $\mathbf{X}_k$, with $k$ defining the sensors set, i.e., $k \in \mathcal{P} = \{p_1, \ldots, p_P\}$. All measured data are organized in vector $\mathbf{U}^m$. On the other hand, the computed displacements at the sensors locations are grouped in vector $\tilde{\mathbf{U}}$, a simple extraction from the nodal displacement $\mathbf{U}$, with $\tilde{\mathbf{U}} \subset \mathbf{U}$.

The model's validation implies that, for a set of applied forces $\{\mathbf{F}_1, \ldots, \mathbf{F}_S\}$, the corresponding computed nodal displacement $\{\mathbf{U}_1, \ldots, \mathbf{U}_S\}$, from which the predicted displacements at the sensors locations are extracted leading to $\{\tilde{\mathbf{U}}_1, \ldots, \tilde{\mathbf{U}}_S\}$, are assumed byy verifying the following:

$$\max_{k=1,\ldots,S} \|\tilde{\mathbf{U}}_k - \mathbf{U}_k^m\|_2 < \epsilon, \tag{11}$$

with $\epsilon$ being a positive coefficient that is small enough.

Thus, we assume that structural model $\mathbf{K}$ is accurate enough for predicting the nodal displacements related to any loading $\mathbf{F}$; that is, at the measurement locations, the predicted and the measured displacements remain close enough. Now, we consider that a deviation is observed, $\|\tilde{\mathbf{U}} - \mathbf{U}^m\|_2 > \epsilon$, and is possibly induced by the structure degradation, and we assume in the first approximation that it is localized (that is, affecting a small region $\omega$ of the entire structure $\Omega$), with $|\omega| \ll |\Omega|$.

In that case, it is expected that both the nominal model and the nominal displacement must correct the former ($\Delta \mathbf{K}$) and update the last ($\Delta \mathbf{U}$); that is:

$$(\mathbf{K} + \Delta \mathbf{K})(\mathbf{U} + \Delta \mathbf{U}) = \mathbf{F}, \tag{12}$$

and taking into account that $\mathbf{KU} = \mathbf{F}$, the previous system reduces to the following.

$$\mathbf{K}\Delta\mathbf{U} + \Delta\mathbf{K}\mathbf{U} + \Delta\mathbf{K}\Delta\mathbf{U} = \mathbf{0}. \tag{13}$$

The linearized problem reads as follows:

$$\mathbf{K}\Delta\mathbf{U} + \Delta\mathbf{K}\mathbf{U} \approx \mathbf{0}, \tag{14}$$

and it requires a parametrization of $\Delta\mathbf{K}$. A possible approximation of it reads as follows:

$$\Delta\mathbf{K} = \sum_{e=1}^{\mathrm{E}} a_e \mathbf{K}_e, \tag{15}$$

where $\mathrm{E}$ is the number of elements involved in the mesh that covers domain $\Omega$, i.e., $\Omega = \cup_{e=1}^{E}\Omega_e$, where $\mathbf{K}_e$ is the nominal stiffness matrix related to element $\Omega_e$ expressed in the global nodal numbering. Expression (15), with $a_e < 0$, represents a simple approximate method of diminishing the contribution of element $\Omega_e$ to the structure's stiffness matrix $\mathbf{K}$. For elements $\Omega_{e'}$ located in the undamaged region $\Omega - \omega$, $\Omega_{e'} \subset \{\Omega - \omega\}$, there is no need for correction, and the associated coefficients will vanish, i.e., $a_{e'} = 0$.

Thus, it is expected that only the elements in the set $\mathcal{E}$ concerned by the elements covering the damaged area $\omega$ ($\omega = \cup_{e \in \mathcal{E}} \Omega_e$) will have non zero values of the $a$-coefficients, while the largest majority of the others (the ones concerning elements in the undamaged region $\Omega - \omega$) will vanish.

Therefore, the minimization problem that enforces the expected sparsity is described as follows.

$$\begin{cases} \min_{a_1,\ldots,a_E} \sum_{e=1}^{\mathrm{E}} |a_e| \\ \text{subjected to} : \mathbf{K}\Delta\mathbf{U} + \Delta\mathbf{K}\mathbf{U} = \mathbf{0} \ \& \ \left(\tilde{\mathbf{U}} + \widetilde{\Delta\mathbf{U}}\right) = \mathbf{U}^m \end{cases}. \tag{16}$$

Because of the linearized model employed, the solution cannot be good enough. On the other hand, trying to iterate by particularizing the nonlinear term at the previous iteration (as most usual linearization strategies consider) has, as a main drawback, the accumulative increase in the corrected elements with the iterations of the nonlinear optimization problem. To circumvent these difficulties, we consider the following iterative procedure (Algorithm 1):

---

**Algorithm 1:** Sparse identification

---

1 At first iteration, $q = 1$, solve the minimization problem (16) :
$$\begin{cases} \min_{a_1,\ldots,a_E} \sum_{e=1}^{\mathrm{E}} |a_e| \\ \text{subjected to} : \mathbf{K}\Delta\mathbf{U} + \Delta\mathbf{K}\mathbf{U} = \mathbf{0} \ \& \ \left(\tilde{\mathbf{U}} + \widetilde{\Delta\mathbf{U}}\right) = \mathbf{U}^m \end{cases}$$
2 Group, from the computed solution, all the elements with non-zero coefficients $a_e^q \neq 0$, in the set $\tilde{\mathcal{E}}^q$: $\tilde{\mathcal{E}}^q = \{e \in \{1,\ldots,\mathrm{E}\}$ such that $a_e^q > \epsilon\}$;
3 Extract a domain $\omega^q$ that covers all the elements in $\tilde{\mathcal{E}}^q$;
4 Re-distribute the sensors, in $\omega^q$ (at this step, a reduction in the number of sensor is also possible) (This step is not essential in the resolution of the problem, it's just used to optimize the algorithm)
5 Solve the minimization problem (16) but by restricting the candidate elements to be corrected to the ones concerned by $\omega^q$, referred by $\mathcal{E}^q$, with $\tilde{\mathcal{E}}^q \subset \mathcal{E}^q$, i.e.,:
$$\begin{cases} \min_{\mathbf{a}^q \in \mathcal{E}^q} \sum_{e \in \mathcal{E}^q} |a_e^q| \\ \text{subjected to} : \mathbf{K}\Delta\mathbf{U} + \Delta\mathbf{K}\mathbf{U} = \mathbf{0} \ \& \ \left(\tilde{\mathbf{U}} + \widetilde{\Delta\mathbf{U}}\right) = \mathbf{U}^m \end{cases}$$
6 Go back to step 2 with $q \leftarrow q + 1$

---

The converged solution allows computing both the model's local correction $\Delta\mathbf{K}$ and the data's completion $\mathbf{U} + \Delta\mathbf{U}$ for which its extraction at the sensor's location $\widetilde{\mathbf{U} + \Delta\mathbf{U}}$ approaches measures $\mathbf{U}^m$.

**Remark 1.** *In order to enhance the localization capabilities by enabling the most accurate correction,* A *different loadings can be applied and are associated with* $^a\mathbf{U}$ *and* $^a\mathbf{U}^m$, $a = 1, \ldots,$ A.

### 3.2. Data Completion and Model Global Enrichment

The problem becomes more complex when $\mathbf{K}$ represents an approximation of the real behaviour almost everywhere. Thus, from a reasonable amount of data $\mathbf{U}^m$, one could be interested in improving the model's quality by adding model enrichment $\Delta\mathbf{K}$; however, if $\mathbf{K}$ must be enriched everywhere, all coefficients $a_e$, $e = 1, \ldots,$ E, are expected being non-null, i.e., $a_e \neq 0$, $e = 1, \ldots,$ E, and, consequently, the problem cannot be casted into a sparse identification problem. Therefore, the simplest approach in order to take into account that the enrichment is expected to be applied almost everywhere (sparsity is not enforced any more) is to replace the L1-norm in problem (16) by the L2-norm. In this case, the enrichment problem is as follows.

$$\begin{cases} \min_{a_1, \ldots, a_E} \sum_{e=1}^{E} |a_e|^2 \\ \text{subjected to} : \mathbf{K}\Delta\mathbf{U} + \Delta\mathbf{K}\mathbf{U} = \mathbf{0} \ \& \ \left(\tilde{\mathbf{U}} + \widetilde{\Delta\mathbf{U}}\right) = \mathbf{U}^m \end{cases}. \tag{17}$$

## 4. Numerical Results

### 4.1. Localized Behaviour: Plate

In this section, we consider the structural model illustrated in Figure 1, which consists of a plate of dimensions $L_x = 1$ m, $L_y = 2$ m and $L_z = 0.1$ m that is fully clamped on its left boundary, free on the other boundaries and subjected to a uniform force distribution on its right side $\mathbf{T} = (0.4 \times 10^5, 0)$ Pa. The plate's behaviour is assumed to be linearly elastic, with the Young modulus is $E = 2$ GPa and the Poison coefficient is $\nu = 0.3$.

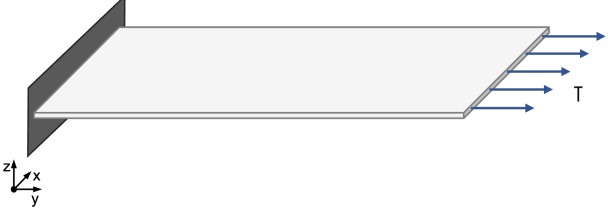

**Figure 1.** Structural model considered for illustrating the local model correction methodology.

This configuration corresponds to the nominal structural system, whereas the real system is assumed to contain a damaged area. The plate is equipped with a finite element mesh, depicted in Figure 2, which serves as support for the approximation of the different mechanical fields: displacement, strains and stresses. The elements located in the damaged region are highlighted in red in Figure 2; here, Young's modulus is reduced to $E' = 0.1 \cdot E$. The sensors location is assumed, for the sake of simplicity, to coincide with some nodes of the mesh, particularly the blue ones in Figure 2.

After solving the minimization problem (16), the elements in which $|a_e| > \epsilon = 0.7 \min\{a_e\}$ are identified and highlighted in red in Figure 3. It can be observed that the region in which the damage is located in is identified; however, the exact damage distribution remains poorly represented.

With both the identified damaged elements and the identified reduced mechanical properties (reduced element stiffness), the solution of the mechanical problem was solved and compared with the reference solution, as depicted in Figures 4 and 5. The proposed procedure exhibits an excellent accuracy for correcting the model while completing the data.

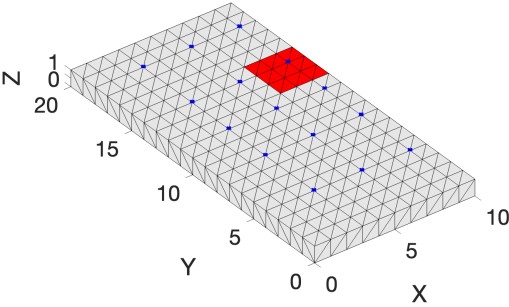

**Figure 2.** Damaged region (red) and location of the 30 sensors (nodes highlighted in blue).

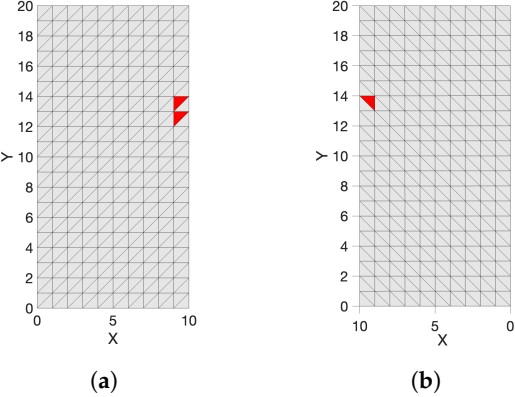

(**a**)           (**b**)

**Figure 3.** Identified elements affected by the damage (in red) at the first iteration ($q = 1$) of the algorithm proposed in Section 3.1. (**a**) Top view; (**b**) bottom view.

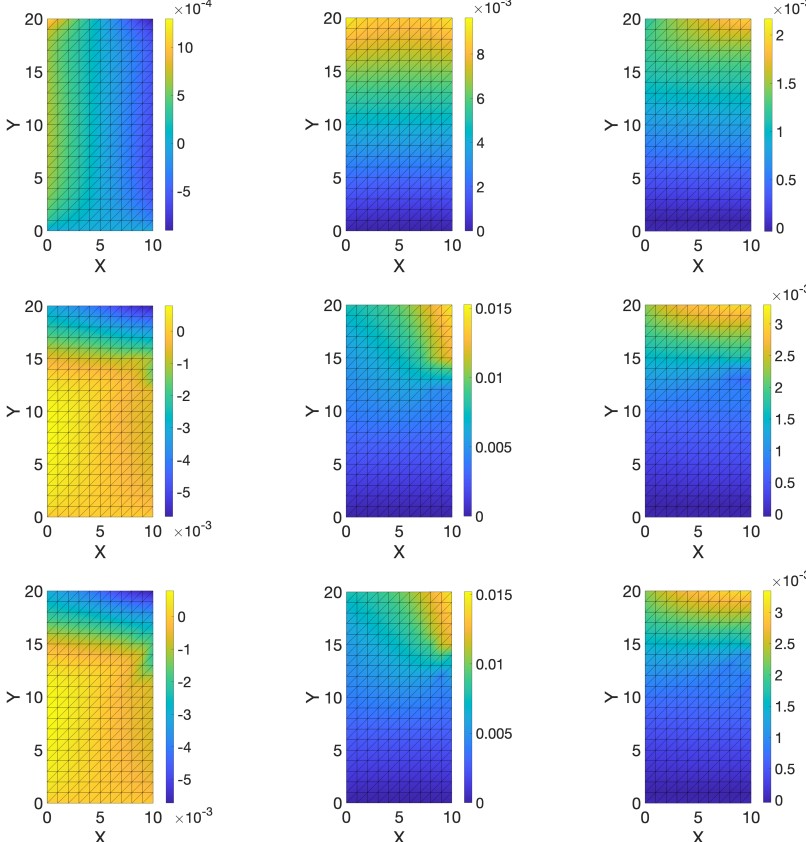

**Figure 4.** Top view of $x$ (**left**), $y$ (**center**) and $z$ (**right**) components of the displacement field associated with the nominal solution (**top**); reference solution that takes into account the real damaged region (**middle**) and corrected model solution (**bottom**).

The errors between the corrected and the reference (the nominal and the reference) displacements field, for $x$, $y$ and $z$, components are as follows.

- $\left\| U_x^{Corr} - U_x^{Ref} \right\|_2 = 2.2 \times 10^{-4} \; (3.3 \times 10^{-2})$ , $err_\%^x = \dfrac{\left\| U_x^{Corr} - U_x^{Ref} \right\|_2}{\left\| U_x^{Ref} \right\|_2} \times 100 = 6.4\% \; (98.2\%)$

- $\left\| U_y^{Corr} - U_y^{Ref} \right\|_2 = 3.8 \times 10^{-4} \; (3.3 \times 10^{-2})$ , $err_\%^y = \dfrac{\left\| U_y^{Corr} - U_y^{Ref} \right\|_2}{\left\| U_y^{Ref} \right\|_2} \times 100 = 2.9\% \; (26.1\%)$

- $\left\| U_z^{Corr} - U_z^{Ref} \right\|_2 = 9.3 \times 10^{-5} \; (9.6 \times 10^{-3})$ , $err_\%^z = \dfrac{\left\| U_z^{Corr} - U_z^{Ref} \right\|_2}{\left\| U_z^{Ref} \right\|_2} \times 100 = 3.1\% \; (32.6\%)$

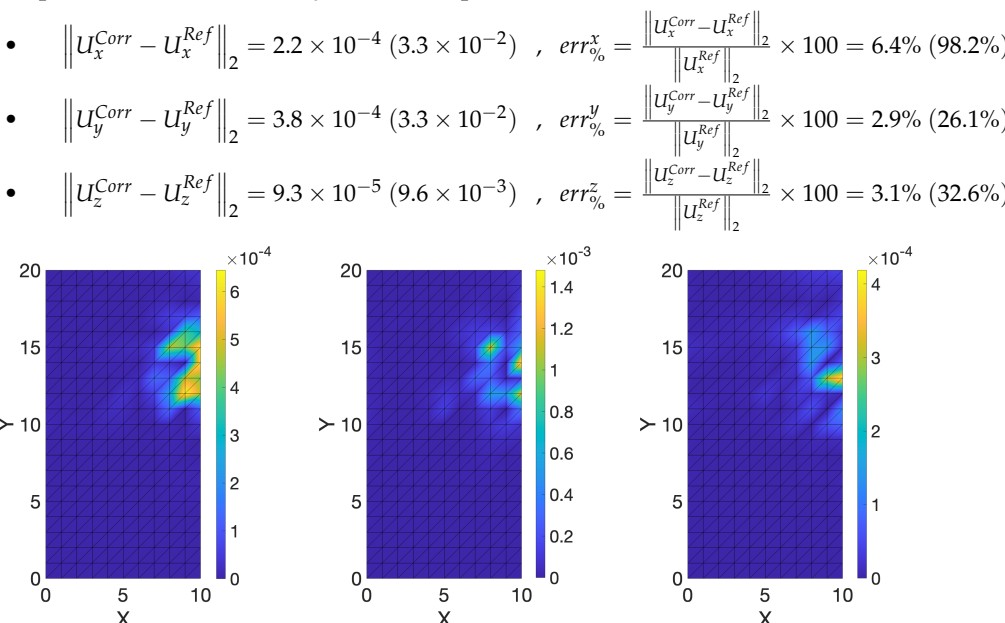

**Figure 5.** Top view of the difference in absolute value between the corrected and the reference displacement field, for $x$ (**left**), $y$ (**middle**) and $z$ (**right**) components.

Of course the quality of this correction technique is highly related to the number and the location of the sensors. We performed a study on the relative errors between the reference displacement solution and the one computed from the corrected model for a different number of sensors placed on the structure, as shown in Figure 6. As expected, the relative error for the different components of the displacement field $\left( \left\| U_{x,y,z}^{Corr} - U_{x,y,z}^{Ref} \right\|_2 / \left\| U_{x,y,z}^{Ref} \right\|_2 \right)$ decreases when the number of sensors increases, as Figure 7 proves.

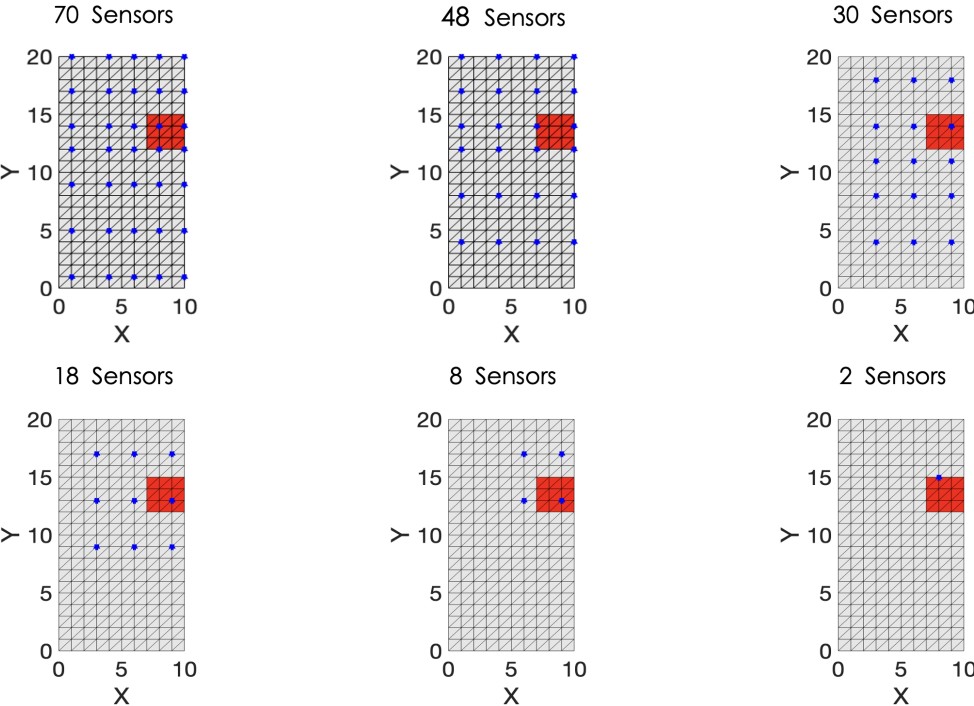

**Figure 6.** Top view of sensors location for different configurations.

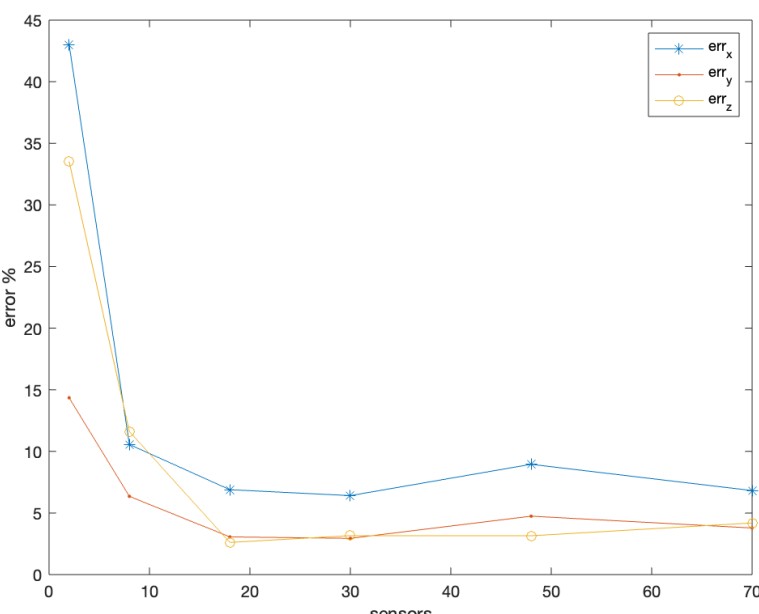

**Figure 7.** Representation of the relative error $\left( \left\| U_{x,y,z}^{Corr} - U_{x,y,z}^{Ref} \right\|_2 / \left\| U_{x,y,z}^{Ref} \right\|_2 \right)$ in percentage, with respect to the number of sensors for the three direction (blue for x-direction, red for y-direction and yellow for the z-direction). In particular, the result for the 6 different configurations (Figure 6) are specified with some marker and connected with straight lines.

The main drawback related to this methodology is the fact that, at the first iteration of the procedure described in Section 3.1, the corrected matrix parametrization considers all elements of the mesh. Thus, when the problem is large (the mesh is too refined), the minimization problem to solve (16) becomes computationally very expensive. To avoid such problems, one could consider a coarser parametrization of the stiffness matrix correction involving larger subdomains where the correction is eventually performed, and then it is refined by zooming in the identified region and so on until reaching an accurate enough localization.

### 4.2. Localized Behaviour: Plate with Hole

In this section, we consider the structural model illustrated in Figure 8 consisting of a plate of dimensions $L_x = 1.5$ m, $L_y = 3$ m and $L_z = 0.15$ m, with a hole in the middle of radius $R = 0.5$ m fully clamped on its left boundary, free on the other boundaries and subjected to a uniform force distribution on its right side $\mathbf{T} = (2 \times 10^3, 0, 0)$ Pa . The plate's behaviour is assumed to be linearly elastic, with a Young's modulus of $E = 2$ MPa and a Poison coefficient of $\nu = 0.3$.

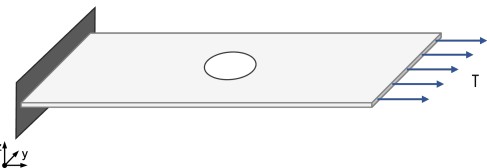

**Figure 8.** Schematic representation of the problem.

This configuration corresponds to the nominal structural system, whereas the real system is assumed to contain a damaged area. The plate is equipped with a finite element mesh, depicted in Figure 9, which serves as a support for the approximation of different mechanical fields: displacement, strains and stresses. The elements located in the damaged region are highlighted in red in Figure 9; here, the Young's modulus is reduced to $E' = 0.1 \cdot E$. The sensor's location is assumed, for the sake of simplicity, to coincide with some nodes of the mesh, particularly the blue ones in Figure 9.

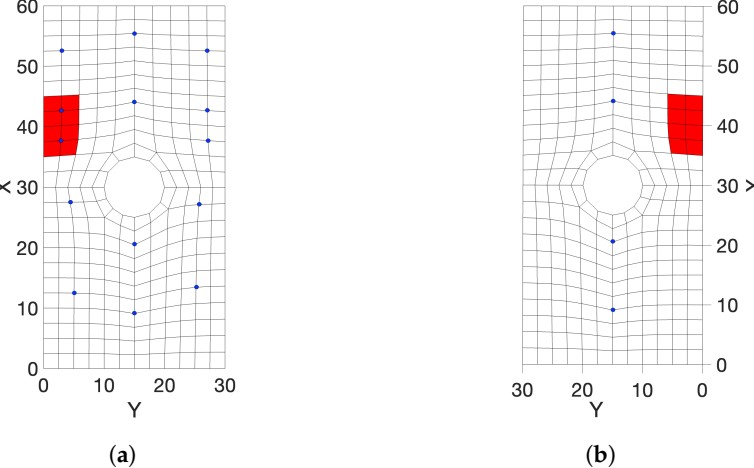

**(a)**　　　　　　　　　　　　　　　　**(b)**

**Figure 9.** Damaged region (red) and location of the 18 sensors (nodes highlighted in blue). (**a**) Top view; (**b**) bottom view.

After solving the minimization problem (16), the elements in which $|a_e| > \epsilon = 0.25 \min\{a_e\}$ are identified and highlighted in red in Figure 10. It can be observed that the region in which the damage is located in is identified; however, the exact damage distribution remains poorly represented.

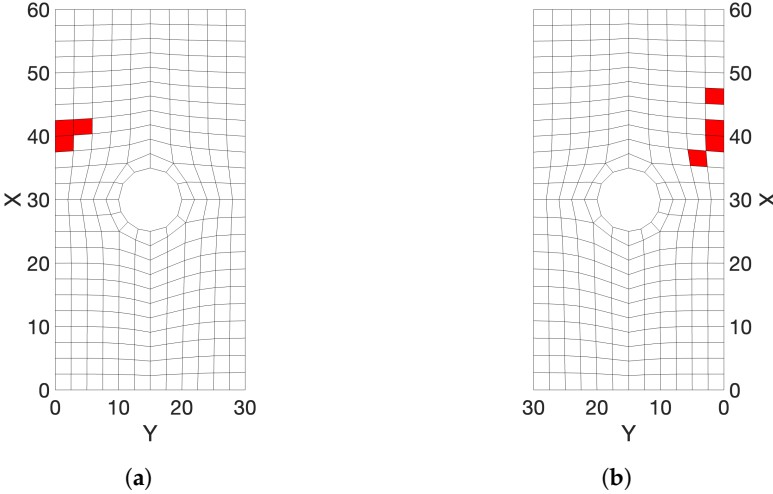

**(a)**　　　　　　　　　　　　　　　　**(b)**

**Figure 10.** Identified elements affected by the damage (in red) at the first iteration ($q = 1$) of the algorithm proposed in Section 3.1. (**a**) Top view; (**b**) bottom view.

With both the identified damaged elements and the identified reduced mechanical properties (reduced element stiffness), the solution of the mechanical problem was solved and compared with the reference solution, as depicted in Figures 11 and 12. The proposed procedure exhibits an excellent accuracy for correcting the model while completing the data.

The errors between the corrected and the reference (the nominal and the reference) displacements field, for $x$, $y$ and $z$ components, are as follows:

- $\left\| U_x^{Corr} - U_x^{Ref} \right\|_2 = 4.3 \times 10^{-3} \ (2.6 \times 10^{-2})$ , $err_\% = \dfrac{\left\| U_x^{Corr} - U_x^{Ref} \right\|_2}{\left\| U_x^{Ref} \right\|_2} \times 100 = 3.1\% \ (19.4\%)$;

- $\left\| U_y^{Corr} - U_y^{Ref} \right\|_2 = 2.5 \times 10^{-3} \ (2.1 \times 10^{-2})$ , $err_\% = \dfrac{\left\| U_y^{Corr} - U_y^{Ref} \right\|_2}{\left\| U_y^{Ref} \right\|_2} \times 100 = 9.6\% \ (85.1\%)$;

- $\left\| U_z^{Corr} - U_z^{Ref} \right\|_2 = 3.7 \times 10^{-3} \ (5.6 \times 10^{-4})$ , $err_\% = \dfrac{\left\| U_z^{Corr} - U_z^{Ref} \right\|_2}{\left\| U_z^{Ref} \right\|_2} \times 100 = 49.7\% \ (7.5\%)$.

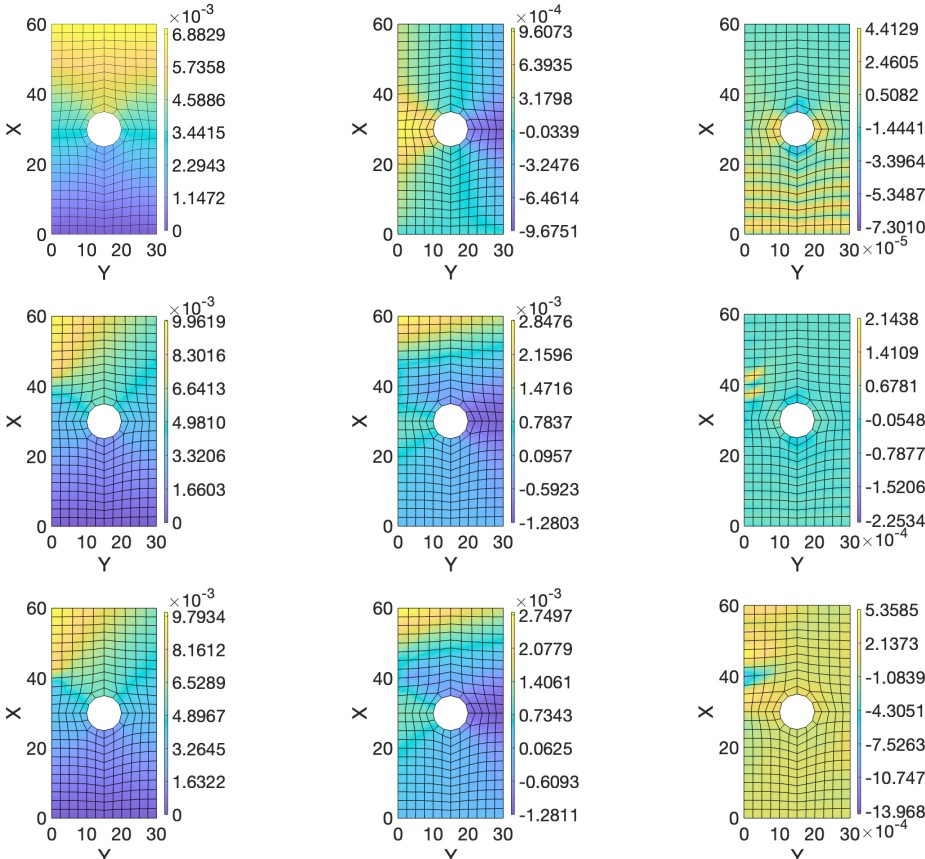

**Figure 11.** Top view of *x* (**left**), *y* (**center**) and *z* (**right**) components of the displacement field associated with the nominal solution (**top**); reference solution that takes into account the real damaged region (**middle**) and corrected model solution (**bottom**).

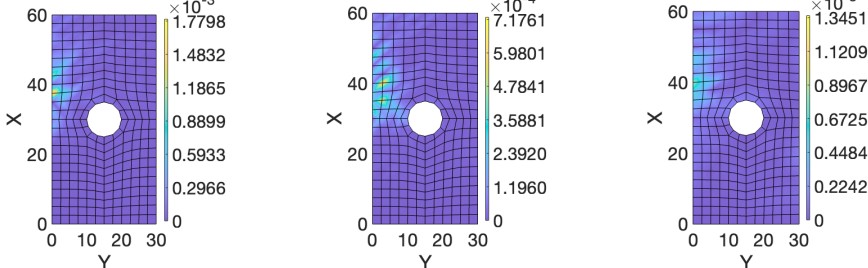

**Figure 12.** Top view of the difference in absolute value between the corrected and the reference displacement field, for *x* (**left**), *y* (**middle**) and *z* (**right**) components.

### 4.3. Localized Behaviour: L-Shape Structure

In this section, we consider a L-shaped structure illustrated in Figure 13. The structure is fully clamped on its upper boundary, free on the other boundaries and subjected to a uniform force distribution on its right side $\mathbf{T} = (0, 20, 0)$ Pa. The plate's behaviour is assumed to be linearly elastic, with a Young's modulus of $E = 200$ Pa and a Poison coefficient of $\nu = 0.3$.

This configuration corresponds to the nominal structural system, whereas the real system is assumed to contain a damaged area. The structure is equipped with a finite element mesh, depicted in Figure 14, that serves as support for the approximation of the different mechanical fields: displacement, strains and stresses. The elements located in the damaged region are highlighted in red in Figure 14; here, the Young's modulus is reduced to $E' = 0.1 \cdot E$. The sensors location is assumed, for the sake of simplicity, to coincide with some nodes of the mesh, particularly the blue ones in Figure 14.

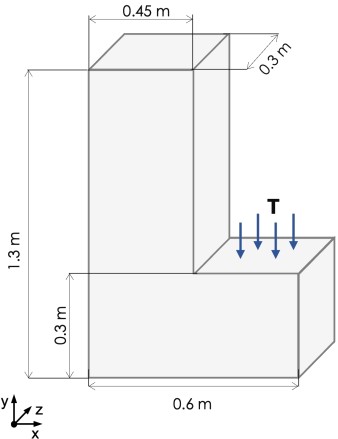

**Figure 13.** Schematic representation of the problem.

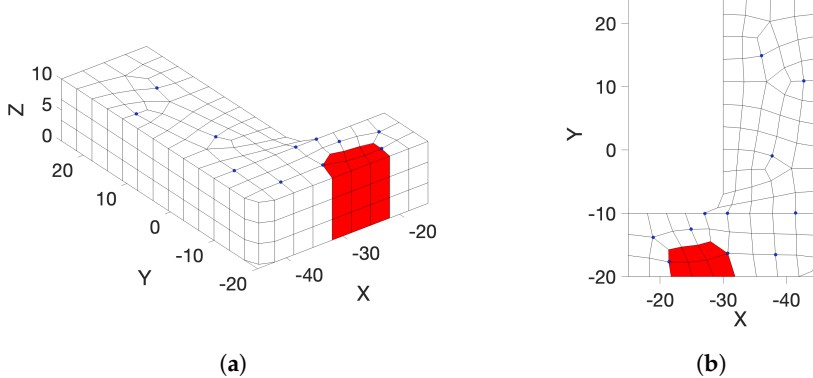

(**a**)                                                                      (**b**)

**Figure 14.** Damaged region (red) and location of the 22 sensors (nodes highlighted in blue). (**a**) Top view; (**b**) bottom view.

After solving the minimization problem (16), the elements in which $|a_e| > \epsilon = 0.19 \, \mathtt{min}\{a_e\}$ are identified and highlighted in red in Figure 15. It can be observed that the region in which the damage locates is well identified.

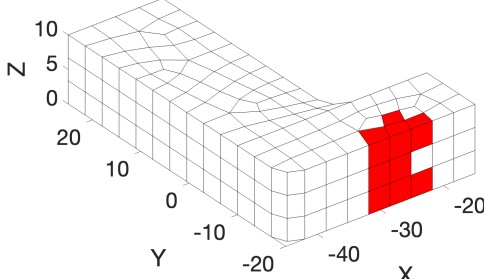

**Figure 15.** Identified elements affected by the damage (in red) at the first iteration ($q = 1$) of the algorithm proposed in Section 3.1.

With both the identified damaged elements and the identified reduced mechanical properties (reduced element stiffness), the solution of the mechanical problem was solved and compared with the reference solution as depicted in Figures 16 and 17. The proposed procedure exhibits an excellent accuracy for correcting the model while completing the data.

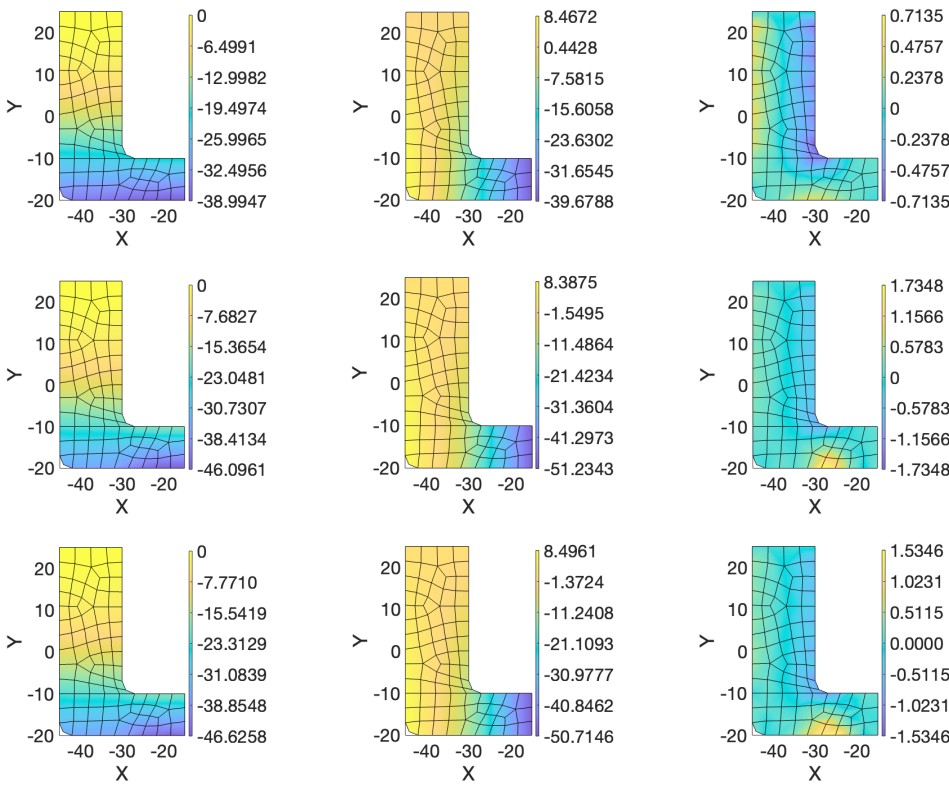

**Figure 16.** Top view of $x$ (**left**), $y$ (**center**) and $z$ (**right**) components of the displacement field associated with the nominal solution (**top**); reference solution that takes into account the real damaged region (**middle**) and corrected model solution (**bottom**).

The errors between the corrected and the reference (the nominal and the reference) displacements field, for $x$, $y$ and $z$ components, are as follows:

- $\left\| U_x^{Corr} - U_x^{Ref} \right\|_2 = 9.36\ (35.6)$ , $err_\%^x = \dfrac{\left\| U_x^{Corr} - U_x^{Ref} \right\|_2}{\left\| U_x^{Ref} \right\|_2} \times 100 = 2.0\%\ (7.9\%)$;

- $\left\| U_y^{Corr} - U_y^{Ref} \right\|_2 = 4.72\ (63.3)$ , $err_\%^y = \dfrac{\left\| U_y^{Corr} - U_y^{Ref} \right\|_2}{\left\| U_y^{Ref} \right\|_2} \times 100 = 1.3\%\ (18.6\%)$;

- $\left\| U_z^{Corr} - U_z^{Ref} \right\|_2 = 1.75\ (3.62)$ , $err_\%^z = \dfrac{\left\| U_z^{Corr} - U_z^{Ref} \right\|_2}{\left\| U_z^{Ref} \right\|_2} \times 100 = 29.7\%\ (61.4\%)$.

**Figure 17.** Top view of the difference in absolute value between the corrected and the reference displacement field, for $x$ (**left**), $y$ (**middle**) and $z$ (**right**) components.

### 4.4. Non-Localized Behaviour: Plate

Now, we consider the same structural model previously addressed (Figure 1). We assume that the structural response of the real plate is described by a linear elastic behavior characterized by a Young's modulus of $E^{\texttt{real}} = 120$ MPa, whereas the nominal model assumes a higher modulus $E^{\texttt{nominal}} = 200$ MPa, with the applied traction $T = 10$ KPa. Thirty sensors are placed at the locations shown in Figure 18.

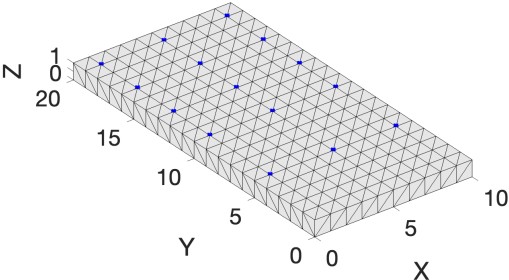

**Figure 18.** Sensors location expected allowing global model enrichment.

The model's enrichment described in Section 3.2 is now applied. The solutions (concerning the *x*, *y* and *z* components of the displacement field) for the reference, the nominal and the enriched models are compared in Figure 19.

In Figure 20, the difference in absolute value between the reference displacement field and the one obtained with the enriched model is shown. These results prove the robustness and ability of the proposed procedure for enriching models while completing the collected data.

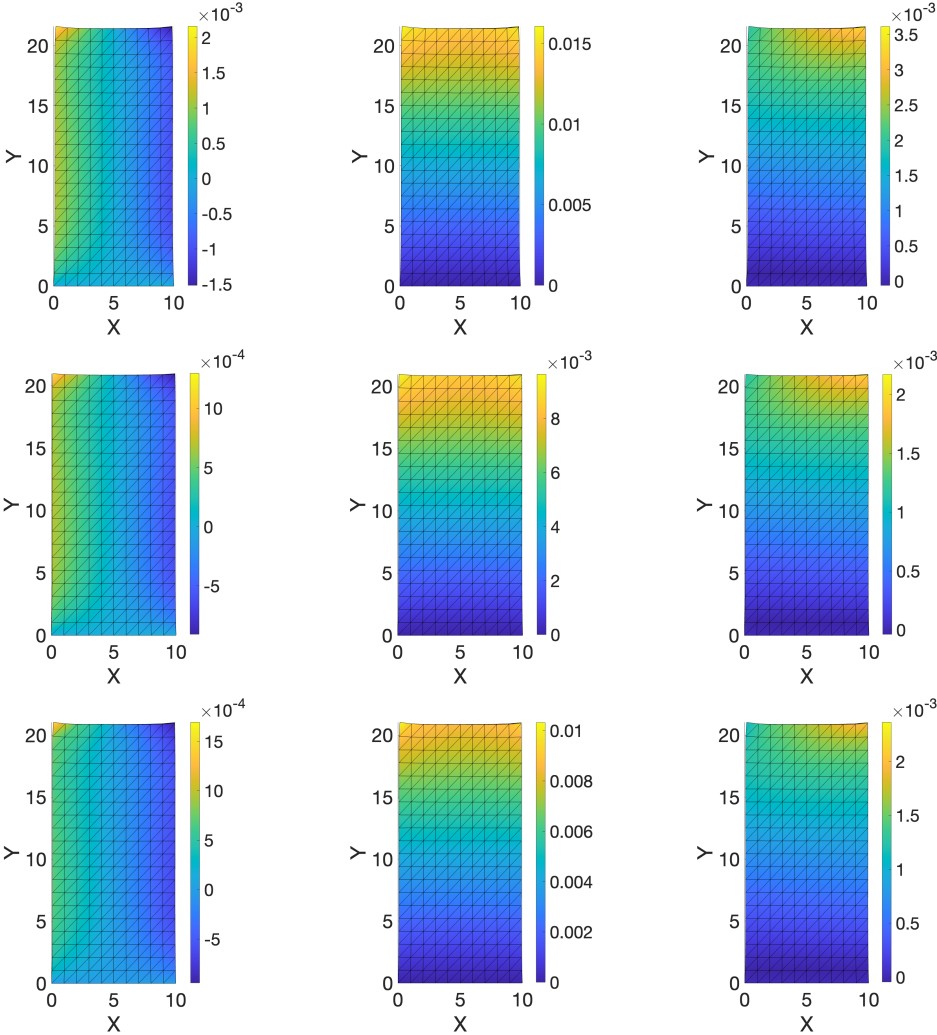

**Figure 19.** Top view of the *x* (**left**), *y* (**center**) and *z* (**right**) components of the displacement field associated to the nominal solution (**top**), the reference solution (**middle**) and the enriched model solution (**bottom**).

The errors between the corrected and the reference (the nominal and the reference) displacements field, for $x$, $y$ and $z$ components, are as follows:

- $\left\|U_x^{Enr} - U_x^{Ref}\right\|_2 = 7.1 \times 10^{-4} \ (5.7 \times 10^{-3})$ , $err_\%^x = \dfrac{\left\|U_x^{Enr} - U_x^{Ref}\right\|_2}{\left\|U_x^{Ref}\right\|_2} \times 100 = 8.4\% \ (66.7\%)$;

- $\left\|U_y^{Enr} - U_y^{Ref}\right\|_2 = 1.5 \times 10^{-3} \ (7.2 \times 10^{-2})$ , $err_\%^y = \dfrac{\left\|U_y^{Enr} - U_y^{Ref}\right\|_2}{\left\|U_y^{Ref}\right\|_2} \times 100 = 1.3\% \ (66.7\%)$;

- $\left\|U_z^{Enr} - U_z^{Ref}\right\|_2 = 3.4 \times 10^{-4} \ (1.4 \times 10^{-2})$ , $err_\%^z = \dfrac{\left\|U_z^{Enr} - U_z^{Ref}\right\|_2}{\left\|U_z^{Ref}\right\|_2} \times 100 = 3.4\% \ (66.7\%)$.

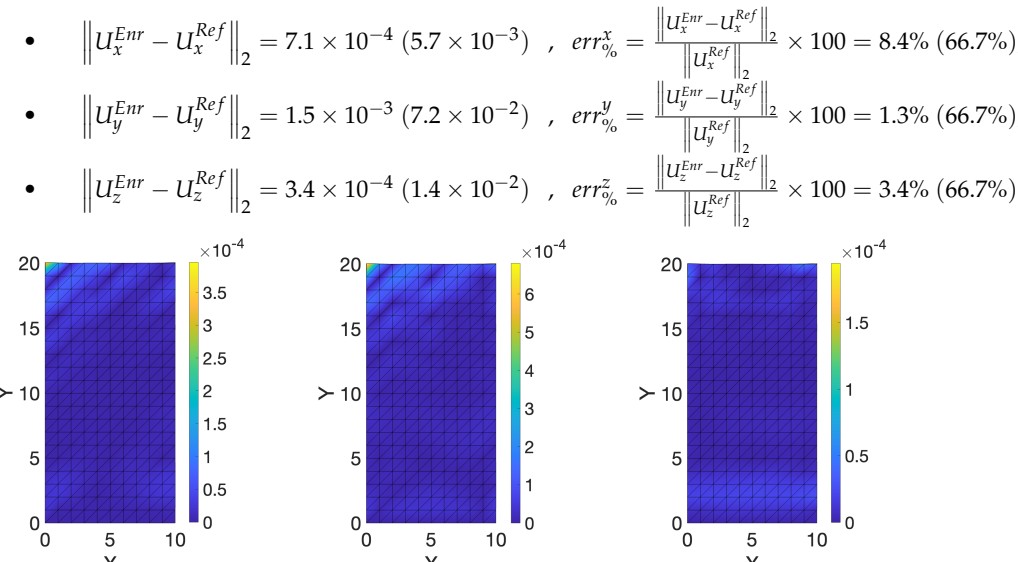

**Figure 20.** Top view of the difference in absolute value between the corrected and the reference displacement field, for $x$ (**left**), $y$ (**middle**) and $z$ (**right**) components.

Again in the present case, the quality of the enrichment is strongly dependent of the number and the location of the employed sensors. We performed a study of the relative error between the reference displacement field and the one computed with the enriched model for a different number of sensors placed on the structure, as shown in Figure 21). As expected, the relative error for each displacement component $\left(\left\|U_{x,y,z}^{Enr} - U_{x,y,z}^{Ref}\right\|_2 / \left\|U_{x,y,z}^{Ref}\right\|_2\right)$ decreases when the number of sensor increases, as Figure 22 proves.

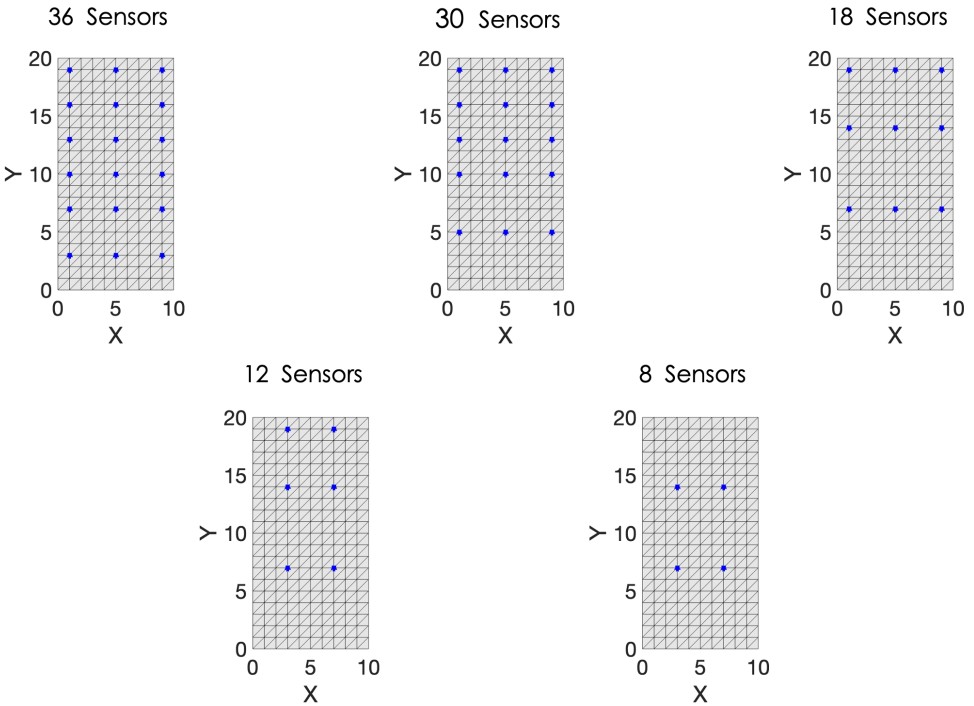

**Figure 21.** Top view of sensors location for different configurations.

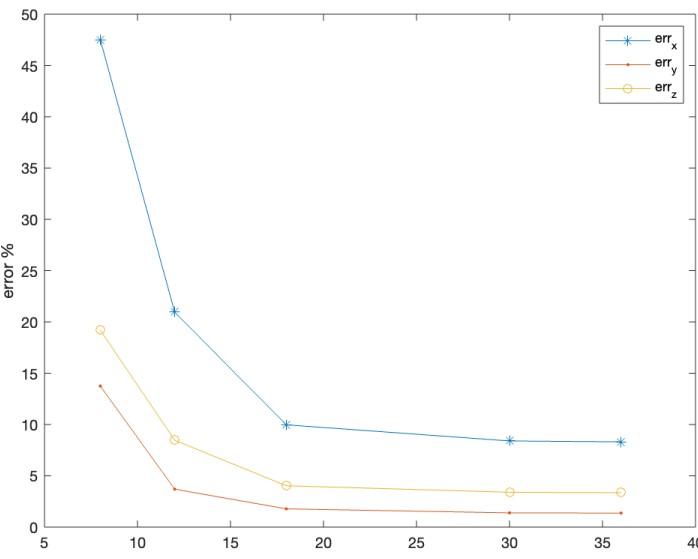

**Figure 22.** Representation of the relative error $\left( \left\| U_{x,y,z}^{Corr} - U_{x,y,z}^{Ref} \right\|_2 / \left\| U_{x,y,z}^{Ref} \right\|_2 \right)$ in percentage, with respect to the number of sensors for the three direction (blue for x-direction, red for y-direction and in yellow for the z-direction). In particular, the result for the 5 different configurations (Figure 21) is specified with some marker and connected with straight lines.

## 5. Conclusions

The present paper proposed two different methodologies for locally correcting or globally enriching models while exhibiting capabilities for completing the data from a few sensors to the entire structural domain. The main aims are as follows: for the first procedure, to locate the damaged elements in a structure, while for the second, to retrieve the correct displacement field proceeding form the sensors data. The sparsity of the model's correction was employed in the first case, and the proposed procedure performed quite well. Moreover, this methodology has quite general results since it can be applied to different geometries. In the second case, when aiming for a global enrichment, the problem was formulated in a similar manner, but the L1-norm promoting sparsity was replace by the usual L2-norm, taking into account the fact that the model enrichment is expected when performed all along the structure.

The originality of the introduced methodology relies on the fact that the correction can be exploited to discover, quantitatively and qualitatively, the discrepancy between the real configuration and the model's configuration.

Future works will address more complex scenarios and nonlinear behaviours. Moreover, the numerical test performed proved the validity and potential of the proposed approaches, which should be now validated with experimental data.

**Author Contributions:** Conceptualization, F.C. and V.C.; methodology, F.C., V.C. and D.D.L.; software, D.D.L., C.G. and V.C.; validation, D.D.L., V.C. and C.G.; formal analysis, D.D.L., V.C. and C.G.; writing—original draft preparation, F.C., D.D.L. and V.C.; writing—review and editing, D.D.L., E.C. and V.C.; visualization, D.D.L.; supervision, F.C. and E.C. All authors have read and agreed to the published version of the manuscript.

**Funding:** This research received no external funding.

**Institutional Review Board Statement:** Not applicable.

**Data Availability Statement:** The datasets are available under demand.

**Acknowledgments:** The authors knowledge the contribution and support of the ESI-ENSAM research chair CREATE-ID. This research is also part of the programme DesCartes and is supported by the National Research Foundation, Prime Minister Office, Singapore under its Campus for Research Excellence and Technological Enterprise (CREATE) programme. This project has received funding from the European Union Horizon 2020 research and innovation programme under the Marie Skłodowska-Curie grant agreement No. 956401 (XS-Meta). The support by the ESI Group through the ESI Chair at ENSAM Arts et Metiers Institute of Technology, and through the project 2019-0060 "Simulated Reality" at the University of Zaragoza is also acknowledged. The support of the Spanish Ministry of Science and Innovation, AEI/10.13039/501100011033, through Grant number PID2020-113463RB-C31 and by the Regional Government of Aragon, grant T24-20R, and the European Social Fund, are also gratefully acknowledged.

**Conflicts of Interest:** The authors declare no conflict of interest.

**Abbreviations**

The following abbreviations are used in this manuscript:

| | |
|---|---|
| MOR | Model order reduction; |
| POD | Proper orthogonal decomposition; |
| ROM | Reduced order modeling; |
| PCA | Principal component analysis; |
| SVD | Singular value decomposition; |
| SVM | Support vector machine; |
| SHM | Structural health monitoring; |
| FEM | Finite element method. |

**Appendix A. A Short Review on SHM**

For our purposes, some techniques look for the changes in the structure's damping ratio, the natural frequencies or the modes shapes [19,33]. When considering dynamical responses, the analysis can be carried out in the time or the frequency domains. Frequency domain techniques take advantage of modal analysis. The literature is extremely abundant. They are based on the use of modal informations extracted from input–output measurements by means of the modal analysis methods or from only using output data measured under the ambient excitation [34,35]. Time domain methods avoid issues related to the natural frequencies. In [15–20], the authors identified modal parameters from time domain measurements and used the extracted vibration features and modal properties for detecting damage occurrence and/or location by comparing the identified modal properties with the original values. Wavelet analysis allows identifying changes in modal shapes, facilitating the damage quantification and its spatial location [21–25,36–38].

Reduced Order Modeling (ROM) techniques have been also extensively considered. In [39], the authors applied the proper orthogonal decomposition (POD) to track the structural behavior. In [40], the authors proposed a data-driven methodology for the detection and classification of damage by using multivariate data-driven approaches and PCA. A support vector machine (SVM) was used for damage detection in [41]. In [42], the authors used machine-learning algorithms to generate a classifier that monitors the damage state of the system and a reduced basis method to reduce the computational burden associated with model evaluations. Proper orthogonal decomposition approximations and self-organizing Maps (SOM) are combined to realize a fast mapping from measured quantities in order to propose a data-driven strategy to assist online rapid decision making for an unmanned aerial vehicle that uses sensed data to estimate its structural state [43].

Data Mining techniques are also used in the context of the SHM. In [44], the authors combined data mining (GA), machine learning (PCA) and deep learning (neural networks) techniques in the damage identification context. Concerning Deep Learning techniques, in [45], a smart monitoring of aeronautical composites plates based on electromechanical impedance measurements and artificial neural networks is proposed.

In our former work [46], we proposed a strategy based on the combination of model order reduction, which extracts a reduced basis from undamaged snapshots, that will serve for projecting any measured solution on it with data-mining techniques. When projecting into this reduced basis the measured field, undamaged regions are expected to be better approximated for the ones in which damage occurs. Thus, data-mining strategies could be then used to differentiate both regions (undamaged and damaged). Finally, in order to limit the number of points at which data are collected, the just-described methodology was combined with a data-completion strategy based on the use of dictionary learning.

These techniques allow identifying and quantifying the damage based on the collected data and on the performed training; however, performing time predictions needed in accurate prognosis remains a challenge for techniques based on data analysis.

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
