# Peer review of "Data Completion, Model Correction and Enrichment Based on Sparse Identification and Data Assimilation"

_applsci, doi:10.3390/app12157458_

Round 1
Reviewer 1 Report
The abstract should be improved to clarify the application, method, results and research value.
Reviewer 2 Report
Very good work has been done and I thank the authors for this new correction methodology and it can help to increase the accuracy of simulation responses of large structures with local damages. I believe that the present manuscript has a high potential for publication in Journal and will definitely have a lot of following. However, it is better to focus on the following points to increase the quality of the paper:
1- The literature review should be improved using more papers published recently.
2- there are several typo that should correct, such as line 114, the phrase "he solution" change to "the solution".
3- what is the mean of symbol ? related to the reference of appendix on page 5 and line 138.
4- It is strongly suggested to shown the proposed procedure in a algorithm (steps 1-6 on page and lines 173-183).
5- The proposed methodology is checked by one simple example using finite element simulation, but, it is clear that the mentioned problems are very important when we have a very complicated geometry or complex loading. So, I think that this example is not suitable and it is better to add one more example with complicated geometry.
Reviewer 3 Report
REVIEW
on article
Data completion, models correction and enrichment based on
sparse identification and data assimilation
Daniele Di Lorenzo, Victor Champaney, Claudia Germoso, Elias Cueto,
Francisco Chinesta
SUMMARY
The article submitted for review is devoted to a topical issue. It considers data completion, correction and enrichment of models based on sparse identification and data assimilation. The relevance of the topic is due to the fact that many models that are supposed to predict the response of structural systems cannot effectively achieve this goal for two main reasons. The first is that some structures undergo local damage during operation, which worsens their mechanical characteristics. The second problem is that the nominal model is sometimes too coarse to capture all the structural details, and as a result, the expected predictions deviate from the measurements.
The authors proposed in their article a solution in the form of a methodology for local correction or global enrichment of models based on the collected data, which in turn are performed outside the location of the sensors.
Thus, the study is relevant, has scientific novelty and practical significance. At the same time, there are shortcomings in the article that need to be corrected, they will be discussed below.
COMMENTS
1. The Abstract of the article does not meet the requirements for abstracts. The abstract considers only the problem, consisting of 2 aspects and tells in the form of a narrative what was done. This is unacceptable, since the abstract must fully reflect the content of the article. The Abstract should be provided with a clear statement of the scientific results obtained. Editors strongly encourage authors to use the following style of structured abstracts, but without headings: (1) Background: Place the question addressed in a broad context and highlight the purpose of the study; (2) Methods: Describe briefly the main methods or treatments applied; (3) Results: Summarize the article's main findings; and (4) Conclusions: Indicate the main conclusions or interpretations. The abstract should be an objective representation of the article.
2. The second remark on the Abstract шы that not only a qualitative result is needed, that is, a statement of the fact of what was done, but also a quantitative characteristic of the result obtained, that is, by what percentage did this or that phenomenon that was the subject of the study improve. If the authors claim that they have improved the models, then it should be shown by what amount the model has been improved.
3. The authors presented the "Introduction" by considering only 12 sources. This is very few for such a topic, which was chosen by the authors. A deeper analysis should be carried out on the topic of models and the accuracy of their representation. There is a large amount of literature on this subject. If this is not done, then there is a fear of not establishing the exact level of scientific novelty. Therefore, it is necessary to consider at least 20-25 sources in the "Introduction" section, conduct a detailed literature review and analysis, and as a result, provide clear statements of scientific novelty, practical significance, goal and objectives of the study.
4. In the “Methods” section, the proposals for Lines 92-94 require, firstly, a small preamble, and secondly, it should be clarified: these are the authors’ own conclusions or based on some previous research. That is, it is necessary to justify the choice of your method in more detail or refer to some previous studies, if necessary.
5. Note the lack of a smooth transition between the "Methods" section and section 3, which is called "Data completion and models correction or enrichment". That is, a clear relationship between these sections should be provided. First, the authors present the choice of method, justify it, and not just present it, and then refine the data and correct or enrich the models. That is, this section is already the resultant one.
6. Figure 7 in section 4 needs more clarification, especially regarding the segments between the points. A detailed interpretation and explanation of it should be provided, since at present it does not look quite clear.
7. The same comments can be applied to Figure 12.
8. The absence of the “Discussion” section as such is noteworthy. That is, the authors presented the results of the study, but did not compare them with the results of other authors. This is unacceptable because the level of significance of the study and the quality of the scientific result are determined in comparison with previous studies. It is necessary to give a detailed analysis and provide links to authors who have done similar research. If the authors claim that they have improved and enriched the model, then it should be explained in what quantitative terms the improvement was made and how this compares with the results obtained earlier by other authors.
9. Conclusions are presented very succinctly and should be reworked. The "Conclusion" section should reflect the main qualitative and quantitative results of the study and the prospects for the development of the study in the future. What do the authors claim as the main scientific result and what trajectories are they developing for themselves in the future?
10. The need for some adjustments in the style of presentation and minor corrections in the English language is noteworthy.
Round 2
Reviewer 3 Report
All my comments were taken into account and appropriate corrections were made in the article's text. Tha article looks much better. I recommend the article for publishing.